# Pluripotency and Growth Factors in Early Embryonic Development of Mammals: A Comparative Approach

**DOI:** 10.3390/vetsci8050078

**Published:** 2021-05-04

**Authors:** Lola Llobat

**Affiliations:** Research Group Microbiological Agents Associated with Animal Reproduction (PROVAGINBIO), Department of Animal Production and Health, Veterinary Public Health and Food Science and Technology (PASAPTA) Facultad de Veterinaria, Universidad Cardenal Herrera-CEU, CEU Universities, 46113 Valencia, Spain; maria.llobatbordes@uchceu.es

**Keywords:** embryo, development, gene expression, growth factors, mammalian, molecular mechanisms, pluripotency

## Abstract

The regulation of early events in mammalian embryonic development is a complex process. In the early stages, pluripotency, cellular differentiation, and growth should occur at specific times and these events are regulated by different genes that are expressed at specific times and locations. The genes related to pluripotency and cellular differentiation, and growth factors that determine successful embryonic development are different (or differentially expressed) among mammalian species. Some genes are fundamental for controlling pluripotency in some species but less fundamental in others, for example, *Oct4* is particularly relevant in bovine early embryonic development, whereas *Oct4* inhibition does not affect ovine early embryonic development. In addition, some mechanisms that regulate cellular differentiation do not seem to be clear or evolutionarily conserved. After cellular differentiation, growth factors are relevant in early development, and their effects also differ among species, for example, insulin-like growth factor improves the blastocyst development rate in some species but does not have the same effect in mice. Some growth factors influence genes related to pluripotency, and therefore, their role in early embryo development is not limited to cell growth but could also involve the earliest stages of development. In this review, we summarize the differences among mammalian species regarding the regulation of pluripotency, cellular differentiation, and growth factors in the early stages of embryonic development.

## 1. Introduction

A successful pregnancy is a complex process that depends on different events that occur in an embryo and the maternal environment. At the preimplantation stages, sequential expression of specific genes in the embryo enable it to implant in the maternal endometrium, while failures in their expression or in their regulation cause pregnancy loss. Therefore, there are considerable differences between spatial and temporal transcriptomes and their regulatory pathways. After formation of the totipotent zygote, sequential cellular divisions occur and the morula is developed [1]. During the morula stage (until the 16-cell stage), the embryo is a compact sphere of cells where cell-to-cell tight junctions are first established [2]. Shortly after, the blastocyst stage is reached and two areas are differentiated: the inner cell mass (ICM), which is composed of the pluripotent epiblast (EPI) and hypoblast (HP) cells in the gastrula stage, and the trophectoderm (TE), which later forms the embryonic placenta. Blastocyst formation happens at different time points depending on the mammalian species. In mice, blastocyst formation occurs approximately 3 days post coitum; in rabbits and humans, it occurs around 3–6 days; in swine and horses, at 7 days; and in cattle and goats, around 7–8 days [3,4,5,6,7,8]. Finally, effective implantation takes place over a span of 6 days in pigs and 28 days in horses [9,10]. The precise genetic regulation of all these processes and their underlying molecular mechanisms are not conserved across species [1]. The differences between the time of embryonic development and implantation in mammal species are shown in Figure 1. Although these events occur at similar times in some species, the expression of certain transcription factors and growth factors vary. In other cases, the transcription factors and growth factors that have been found are the same, but their regulatory mechanisms and pathways differ between species. To extend the challenge of establishing pluripotent stem cells in domestic animals, it is necessary to understand the establishment of pluripotency and how growth factors influence early embryos. In this review, we focus on the most important factors that orchestrate early stages in embryonic development and their species-specific gene regulatory patterns.

## 2. Pluripotency Transcription Factors

One of the most critical transcription factors related to pluripotency and regulated by the Wnt canonical pathway in several mammals is the *Oct4* transcription factor (belonging to the POU gene family, POU5F1 gene), which is expressed predominantly in pluripotent cells [14,15]. This transcription factor is necessary to maintain pluripotency, but its presence differs among species. In fact, *Oct4* is expressed both in the ICM and TE in human, mouse, rabbit, pig, sheep, and cattle preimplantation embryos [4,8,16,17,18]; however, early development can be supported without *Oct4* expression in bovines, so maternal *Oct4* mRNA maintains its expression [19]. In goat embryos, the inhibition of *Oct4* does not affect blastocyst formation but does increase the expression of other genes, such as Nanog homeobox (*Nanog*) [20]. The relative expression of *Oct4* remains constant between the oocyte and morula stage, and decreases in blastocyst in in vitro studies, indicating the beginning of cellular differentiation. Factors such as *Nanog* and SRY-box transcription factor 2 (*SOX2*) are upregulated by the Wnt pathway in ICM around Day 8 in bovine embryos, with one (*SOX2*) or two (*Nanog*) characteristic peaks of expression in goat embryos at the 8- to 16-cell stage and later blastocyst stage, respectively. In this context, *Nanog* expression is necessary for the proliferation of TE cells [5,15,21,22,23].

Ozawa et al. [24] analyzed gene expression between the ICM and the TE in bovine embryos, and showed that *Nanog* and *SOX2* presented similar expression patterns in bovine embryos obtained from mice and humans. *Nanog* expression was higher in ICM than TE in bovine expression, while *Oct4* expression was similar, and *Nanog* is necessary for the expression of *SOX2* (marker of EPI cells), *GATA6* (marker of HP cells), and *CDX2* [25]. *Oct4*, *Nanog*, and *SOX2* expression in bovine embryos is regulated by the exogen bone morphogenetic protein 5 (BMP5) [26]. In human embryos, BMP5 and other BMPs regulate different sets of developmental genes, such as *GATA2*, *GATA3*, and *CDX2*, and BMP10 is the one with greater regulatory potential [27]. Recently, Naddafpour et al. [20] reported that *Oct4* inhibition in goat embryos increased the relative expression of *CDX2*. This gene presents an important function in mouse embryos, since, after TE formation, it inhibits *Oct4* expression in their cells and allows the differentiation of TE cells [28]. In fact, *Oct4* is not required to inhibit *CDX2* expression in bovine and human ICM, but is necessary for the expression of *Nanog* [19,29]. The molecular mechanism that maintains *Nanog* expression in the absence of *Oct4* is still unknown in mice. Additionally, *GATA6* expression is repressed by *Oct4* to facilitate HP differentiation [30,31]. This inhibition in mice occurs by blocking the MEK/ERK pathway, which leads to *GATA6* downregulation in HP and depends on the fibroblast growth factor (FGF) [32]. In humans, cattle, pigs, and rabbits, the segregation of HP is independent of FGF [33,34,35]. MEK inhibition reduces the number of HP cells in mice and rats, but not in humans, cattle, pigs, and rabbits [33].

Transcriptomic analyses of embryos from different species have identified additional regulatory factors that modulate pluripotency and cellular differentiation in early development. Bernardo et al. [36] compared pluripotency genes in mouse, pig, and bovine embryos and found that around 82% of the genes were commonly expressed across the three species studied. Regarding *Oct4*, *Nanog,* and *SOX2* expression in the ICM, mice at Day 3.5 showed expression levels similar to those of cattle and pigs at Day 7, and three genes were upregulated between the transition from ICM to EPI in pigs and cattle, while *Nanog* and *SOX2* were downregulated and *Oct4* was stable in mice [33]. For example, bovine embryos upregulated 159 and 48 genes in the ICM and TE, respectively. Genes expressed differentially have been compared using gene ontology with mice and humans, and species-specific pluripotency control in ICM was demonstrated [37]. In pigs, analysis of individual cells by single-cell RNA sequencing showed expression of some species-specific genes, such as paired box 6 (*PAX6*), aquaporin 3 (*AQP3*), and in late blastocyst, clathrin adaptor protein (*DAB2*), platelet-derived growth factor receptor alpha (*PDGFRA*), fibronectin 1 (*FN1*), hepatocyte nuclear factor 4 alpha (*HNF4F*), goosecoid homeobox (*GCS*), nuclear receptor subfamily 5 group A member 2 (*NR5A2*), and lysine acetyl-transferase 6A (*KAT6A*) [38].

In summary, these findings indicate that although the control of pluripotency is mainly carried out through the canonical Wnt/β-catenin and MER/ERK pathways, downstream temporal and spatial development cues differ depending on the species.

## 3. Growth Factors and Early Development

### 3.1. Vascular Endothelial Growth Factor (VEGF)

Adhesion processes can be affected by growth factors that regulate vascularization and cellular motility. One of these factors, vascular endothelial growth factor (VEGF) is associated with de novo vascularization in a wide variety of processes, such as implantation, embryogenesis, menstrual cycle, corpus luteum development, ovarian follicular development, and tumorigenesis [39,40]. Initially, VEGF was characterized for its ability to induce vascularity, permeability, and promote vascular endothelial cell proliferation [41]. Three families of VEGF proteins and their corresponding receptors have been characterized and the main receptors involved in the first steps of signal transduction cascades comprise different tyrosine kinases receptors, such as VEGFR-1, VEGFR-2, and VEGFR-3 [42]. Across species, some VEGF family members and receptors are found in placentomes, uterus tissues, and oviduct, and in different species including humans, mice, rats, cattle, sheep, pigs, and rabbits [43,44,45,46,47,48,49,50]. The in vivo administration of VEGF to goats and sheep stimulates follicular growth and increases the number of preovulatory follicles [51]. Studies in vitro have shown that VEGF supplementation in bovine and pig embryo culture improves cytoplasmatic maturation and blastocyst development rates [52,53]. Recently, Liu et al., (2020) [54] showed that VEGF improved embryo development rates in vitro, on the one hand, through activation of the MAPK pathway and, on the other hand, via inhibition of the canonical Wnt pathway during the last step of oocyte maturation.

In the embryo, the expression of *VEGF* and its receptors (*VEGFR-1* and *VEGFR-2*) are also conserved in different species. For example, *VEGF* expression has been related to fetal weight increase in porcine embryos [55], in which *VEGF* expression was detected in TE cells at Day 14 of pregnancy [50]. *VEGF*, *VEGFR-1*, and *VEGFR-2* mRNA have been found in vitellin sacs and TEs of bovine embryos [49,56,57]. In humans, *VEGF* expression increases in the late luteal phase, while in bovine corpus luteum, *VEGF* mRNA is upregulated in the early luteal phase, and then is progressively downregulated until its levels increase again during pregnancy [58,59]. In rabbits, *VEGF* expression increases around Day 6 of pregnancy before implantation [8].

The role of VEGF in vascularization and fetal growth is known, but additional players are gaining biological relevance in successful embryonic development and implantation in mammals. For instance, a critical role of macrophage recruitment and embryo polarization have been reported [60].

### 3.2. Transforming Growth Factor-Beta (TGF-β) Superfamily

Another relevant group of growth factors that are conserved across species before and during implantation is the transforming growth factor-beta (TGF-β) superfamily. This superfamily comprises regulating factors involved in growth and differentiation, for example, bone morphogenetic proteins (BMPs), activin (Ac), nodal and gonadal hormone growth factors, as well as inhibin (In) [61]. In particular, Ac plays an important role in cellular differentiation, proliferation, and apoptosis [62]. Recently, Bloise et al. [63] published a review and provided an in-depth analyses of the different functions of activin in human reproduction. The authors described the different roles of Ac and highlighted its role promoting endovascular differentiation in TE through VEGF stimulation. In summary, Ac facilitates blastocyst union and TE penetration during the first stages of implantation in humans and mice. These roles of the TGF-β superfamily have also been demonstrated in other species, such as pigs, in which TGF-β regulates blastocyst differentiation and maturation events, including modulating the interaction between the uterus and embryo during implantation [64,65]. Another member of the TGF-β superfamily, growth differentiation factor-8 (DGF-8), is involved in the expression of ICM marker *SOX2* during porcine embryo in in vitro development, indicating its role in preimplantation embryonic development and pluripotency control [66]. In fact, *TGF-β* expression increases in the porcine conceptus–maternal interface at the same time that the embryo is lengthened and the fixation and implantation process begins [67]. In in vitro culture, the addition of TGF-β superfamily factors, such as BMP-15, improves blastocyst development rates in sheep, goats, and cattle [5,68,69]. In fact, expression of TGF-β in early bovine embryos (from two- to eight-cell stages) has been shown to increase the relative abundance of *Nanog*, suggesting an early role of TGF-β [70]. In rabbits, the relative expression of *TGF-β* increases on Day 6 of pregnancy [8]; therefore, this early role of TGF-β could be species specific. The activity of the TGF-β superfamily members is mediated through SMAD signaling in humans and mice, and SMAD2 and SMAD3 are necessary for bovine early embryonic development [71,72]. The AKT pathway seems to play an important role; therefore, SMAD signaling might not be the only way to regulate TGF-β actions in bovine embryos [73].

### 3.3. Fibroblast Growth Factor (FGF) Family

The fibroblast growth factor (FGF) family members are involved in angiogenesis, embryonic development, and have a role in controlling peri-implantation development [74,75,76,77]. Some FGF family members, such as FGF-3 and FGF-8, are related to the differentiation of different tissues in the late stages of development, such as brain, liver, pancreas, and heart, among other organs and tissues in mammals [78,79,80,81,82,83].

In human and mouse embryos, FGF promotes mesoderm formation and proliferation [84,85]. Recently, Guzzeta et al. (2020) demonstrated that a Hedgehog–FGF signaling axis is required for anterior mesoderm lineage development during gastrulation [86]. In mouse blastocyst, the expression of the *FGF-2* receptor is observed, but not in human blastocyst [87]. These data suggest that the expansion of TE in early development depends on FGF in mice, and that this expansion, which is dependent on FGF, occurs more in late stages in humans. The actions of FGF family members begin in the initial steps of reproduction mechanisms, therefore, *FGF-10* expression has been detected in theca cells and ovarian stromal cells in humans [88], and FGF-2 and FGF-10 have been shown to increase the survival and proliferation of cumulus cells, increasing blastocyst development rates in bovine, sheep, and yak, in in vitro cultures [89,90,91,92]. Another family member, FGF-18, seems to regulate steroidogenesis in the fetal ovary [93]. In the early development of bovine embryos, FGF-2 regulates the expression of genes related to the development and proliferation of ICM, such as *Nanog* and *GATA6* [94]. In addition, FGF-1, FGF-2, and FGF-10 produced by TE control the expression of interferon tau (IFN-τ), favoring implantation in cows [75]. FGF family members also play an important role in the early development of pig embryos. FGF-4 regulates TE formation and elongation [95], and the amount of FGF-2 increases in endometrial tissues between Days 15 and 20 of pregnancy, demonstrating its relevance in embryo elongation and implantation [45]. The mechanisms by which FGF family members regulate all these events, and which are the most relevant factors in each species, remain unknown. Table 1 summarizes the more important FGF family members related to reproduction events, localization, and the species in which they are found.

### 3.4. Insulin-Like Growth Factor (IGF) System

Insulin-like growth factors (IGFs) are polypeptides with insulin-like sequences with mitogenic properties that induce proliferation and growth of somatic cells [106]. In addition, fetal and placental growth are regulated by autocrine and paracrine IGFs and their receptors, such as insulin-like growth factor receptor 1 (IGFR-1), insulin-like growth factor receptor 2 (IGFR-2), and insulin receptor (IR) in humans and mice [107]. IGFR-1 is an IR-like tetrameric transmembrane protein with high affinity to IGF-1 and IGF-2 [108,109]. IGFR-2 is a simplex polypeptide with affinity only to IGF-2 [110].

IGF-1 and IGF-2 have been correlated to fetal, placental, and post-partum growth in different species of mammals, including humans, rodents, cattle, sheep, pigs, and dogs [111,112,113,114,115]. In vitro and in vivo studies have shown placental and fetal growth regulation by autocrine and paracrine effect of IGF-1 and IGF-2, and their interactions with IGFR-1, IGFR-2, and IR [107]. These factors and their receptors do not only influence fetal and placental growth but also regulate different signaling cascades to promote cellular proliferation and differentiation in some reproductive steps [116,117]. In fact, *IGF-2*, *IGFR-1*, *IGFR-2*, and *IR* are present in human and mouse oocytes, where an IGF/insulin axis regulates gamete development [118,119]. This regulation differs between species, e.g., rat oocytes only express *IGF-1* but not its receptor *IGFR-1* [120], bovine oocytes express two receptors, *IGFR-1* and *IGFR-2*, and *IGF-1* but not *IGF-2* [121,122], and in dog, corpus luteum presents a similar amount of *IGFR-1*, but expression of *IGF-1* diminishes during embryonic development [123]. In vitro studies have shown that the addition of IGF-1 in culture oocyte medium improves maturation rate and embryonic development in sheep by the PI3/AKT pathways, but the blastocyst development rate is not affected in sheep and mice [124,125,126]. The PI3K/AKT pathway is activated by *IGF-1* and *IGFR-1*, and IR is secreted in early human embryos, whereas this pathway is activated by *IGF-1* in the trophectoderm of horses [127,128]. In fact, *IGF-1* and *IGF-2* are expressed by blastocyst in several mammalian species (Table 2) [48,119,128,129,130,131].

Indeed, the addition of IGF-1 in culture medium activates the PI3K/AKT pathway and improves blastocyst development rates in humans, cattle, goats, yaks, and horses but not in mice and sheep [124,127,128,152,153,154,155]. Similar results have been observed with the addition of IGF-2 in humans and cattle [156,157]. These results indicate that the IGF system is related to placental, fetal, and postnatal growth in mammals and is also involved in the process of cellular differentiation in early embryos. In fact, interactions between *IGF-2* and *IGFR-2* have been shown to regulate homeobox genes controlling apoptosis in human and bovine TE cells [157,158]. Studies in yak early embryos have shown that the addition of IGF-1 increased *Bcl-2* expression (anti-apoptotic gene) and diminished *Bax* expression (apoptotic gene), therefore, this regulation of apoptotic events occurs by different compounds of the IGF system depending on the species [153]. A schematic of the canonical pathway of apoptosis regulation by the IGF system is shown in Figure 2.

Differences in IGF system control such as the fact that the addition of IGF-1 in culture medium improves blastocyst development rates in some species but not in mice, or *IGFR-2* expression in some mammalian blastocysts but not in mice and sheep, indicate relevant alternative pathways of regulation by the IGF system. These results suggest that it would be interesting to use species other than mice, such as bovine or rabbit, for the study of human embryonic development.

### 3.5. Epidermal Growth Factor (EGF) Family

The epidermal growth factor (EGF) family comprises 13 polypeptide proteins that bind to members of the four ErbB family receptors. These four receptors have the ability to bind different growth factors and molecules, and regulate different events such as cell proliferation, migration, differentiation, and apoptosis [159]. Regarding reproductive events, the best-known roles of EGF are in oocyte growth, maturation, and developmental competence in mammals [160]. In cows, da Rosa et al. [161] demonstrated that inhibition of EGF receptors arrested oocyte development in the germinal vesicle stage, and Sugimura et al. [162] determined that EGF was necessary to correct oocyte–cumulus communication. Supplementation of medium with EGF increased oocyte maturation in goats, whereas this improvement in oocyte maturation did not occur in pigs [163,164]. The results in early embryos have been contradictory according to the species and the study. For example, in bovine embryos, in vitro culture supplementation of EGF and insulin-transferrin-sodium selenite (a common complement for the in vitro culture), increased the embryonic development rate and TE capacity of invasion [165]. In contrast with these results, Dall’Acqua et al. [166] did not observe differences in blastocyst development rate in in vitro culture with EGFR inhibitor, and they observed decreased apoptosis in early embryos. These results could indicate that EGF alone does not improve blastocyst development in cattle. Kelly et al. [155] found similar results in sheep; however, EGF seems to improve embryo in vitro production in mice, goats, and pigs [155,163,167,168].

Some studies have demonstrated *EGF* and its receptors’ expression in the early embryos of mice, rabbits, sheep, and pigs, but this expression does not exist in other mammals, such as goats [48,169,170,171,172]. These data suggest that EGF has a role in embryonic development that is species specific, although further studies are necessary to discover this role and its mechanisms.

### 3.6. Other Growth Factors

In addition to the growth factors previously analyzed in this review, other factors that affect early embryonic development and oocyte development have been found in different mammalian species. For example, nerve growth factor (NGF) has a role in embryonic development in sheep and oocyte development in rabbits [173,174]; growth factor receptor-bound protein 10 (GRB10) plays a role in embryonic development in humans and cattle [175,176]; hepatoma-derived growth factor (HDGF) promotes early blastocyst development without bovine serum albumin (BSA) [177]; hepatocyte growth factor activator inhibitor-1 (HAI-1) is necessary for human and murine TE function [178,179]; granulocyte–macrophage colony-stimulating factor (GMCSF) is secreted from cells of the female reproductive tract in mice, accelerating the development of the blastocyst in vitro, and the presence of this growth factor is related to the high proliferation and viability of blastomeres [180]; and platelet-derived growth factor (PDGF) increases the development of bovine and human embryos after the 16-cell stage and morula stage [181,182,183]. In vitro studies have demonstrated an increase in human blastocyst development rate via supplementation with other growth factors such as brain-derived neurotrophic factor (BNF), glial cell-line derived neurotrophic factor (GCLDNF), and colony-stimulating factor (CSF) [183] that reduce oocyte competence and exert a positive effect in *Nanog* and *SOX2* expression in bovine epiblast [184].

In summary, a large number of growth factors are related to embryonic development, and there is still much to be investigated with regard to the roles they play during development in different species of mammals.

## 4. Conclusions

During early embryonic development in mammals, several cellular and molecular mechanisms are activated, each involving different transcription factors and growth factors, related to pluripotency control, and cellular differentiation and growth. Some of these events, however, are species-specific. Therefore, the interaction of these factors with each other, and the metabolic pathways involved remain to be clarified. The use of certain species, such as the mouse, to understand these mechanisms in early pregnancy in humans should be reviewed since substantial differences between the two species are evident. In addition, many questions about regulator genes of pluripotency and cellular differentiation, and other molecules, such as growth factors as well as the interactions among them in different mammalian species, remain to be answered.

## Figures and Tables

**Figure 1 vetsci-08-00078-f001:**
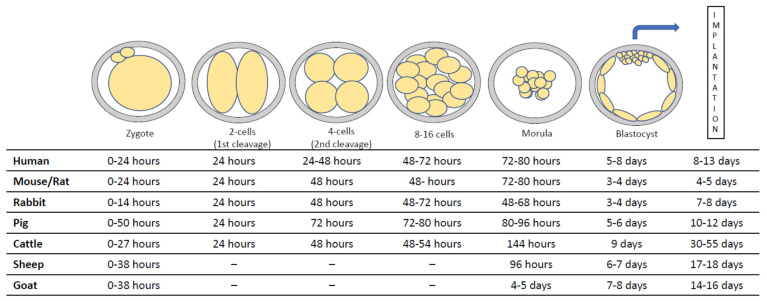
Timing of development after fertilization from a zygote to implantation in different species of mammals [1,11,12,13].

**Figure 2 vetsci-08-00078-f002:**
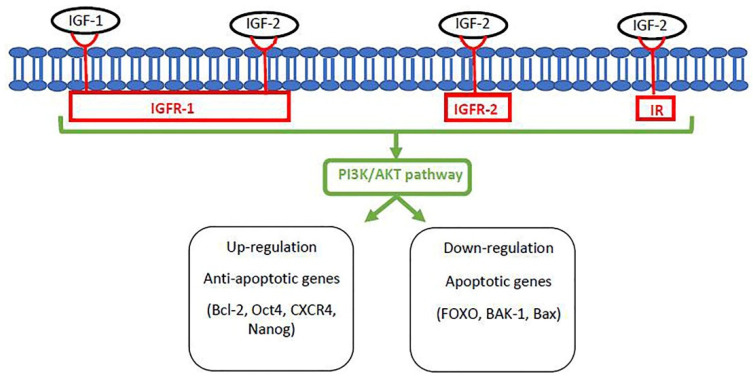
Schematic of the canonical pathway of apoptosis regulation by signaling of the IGF system. *IGF-1* binds to insulin-like growth factor receptor 1 (*IGFR-1*); *IGF-2* binds to *IGFR-1*, *IGFR-2*, and insulin receptor (*IR*), and two factors activate the PI3K/AKT pathway, up- and down-regulating expression of anti-apoptotic and apoptotic genes, respectively.

**Table 1 vetsci-08-00078-t001:** More important fibroblast growth factor (FGF) family members related to reproduction events, localization in embryos, and the species in which they are found.

FGF Family Member	Localization	Species
FGF-1	Mesoderm	Mouse [96]
	Late embryo (Day 12.5)	Mouse [82]
	Trophectoderm	Human [97]
	Mesoderm	Human [98]
	Mesoderm	Rat [99]
	Trophectoderm	Cattle [74]
FGF-2	Trophectoderm	Human [97]
	Mesoderm	Rat [99]
	Trophectoderm	Cattle [74]
	Ectoderm	Pig [100]
	Mesoderm	Pig [100]
	Endoderm	Pig [100]
FGF-4	ICM	Mouse [101]
FGF-6	Somites	Mouse [102]
	Myoblasts	Mouse [103]
FGF-7	Trophectoderm	Cow [104]
FGF-10	Trophectoderm	Cattle [75]
	Teca cells	Human [88]
FGF-18	Late embryo (Day 30)	Human [105]
FGF-23	Late embryo (Day 30)	Human [105]

**Table 2 vetsci-08-00078-t002:** Spatiotemporal expression of insulin-like growth factors (IGFs) across mammalian species.

IGF System Compound	Localization	Species
IGF-1	Blastocyst	Human [132], mouse [133], rat [134], cattle [135], sheep [136], goat [137], rabbit [48], dog [138], buffalo [139]
	Early embryo	Human [140], cattle [129], horse [130]
	Early placenta or pregnant endometrium	Human [141], rat [142], rabbit [48], pig [143], dog [144], horse [127]
IGF-2	Blastocyst	Human [119], mouse [145], rat [142], cattle [146], sheep [137], pig [143], rabbit [48], goat [22], dog [138], horse [130], buffalo [139]
	Early embryo	Cattle [129]
	Early placenta or pregnant endometrium	Human [147], rat [142], rabbit [48], dog [144], cat [148]
IGFR-1	Blastocyst	Human [119], mouse [133], rat [134], cattle [135], sheep [136], rabbit [48], horse [130], cat [149], buffalo [139]
	Early embryo	Cattle [129]
	Early placenta or pregnant endometrium	Dog [144], rabbit [48], rat [134]
IGFR-2	Blastocyst	Human [119], rat [134], cattle [146], pig [143], goat [22], rabbit [48], cat [149], horse [130], buffalo [139]
	Early embryo	Cattle [129]
	Early placenta or pregnant endometrium	Rabbit [48], cat [148]
IR	Blastocyst	Human [119], mouse [150], rat [134], rabbit [151], cattle [150], sheep [136]
	Early embryo	Cattle [129]
	Early placenta or pregnant endometrium	Rat [134]

## Data Availability

Data sharing not applicable.

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
