# Peer review of "Pluripotency and Growth Factors in Early Embryonic Development of Mammals: A Comparative Approach"

_vetsci, 2021, doi:10.3390/vetsci8050078_

Round 1
Reviewer 1 Report
In this review work, the author aimed to summarizes the dissimilarities in the regulation of pluripotency, cellular differentiation, and growth factors in the first phases of embryo development between mammalian species. Overall, data are convincing, but some discrepancies need to be clarified. I have some corrections and comments to the manuscript, which are outlined below.
Introduction
Line 46: what the author means with compact ball? I think there are more appropriate words.
Line 49: please, add the bracket to (HP.
Section 2 - Pluripotency transcription factors
Lines 71-73: which is the reason why Oct4 decrease after the morula stage? Please specify.
Lines 84-86: the AA could differentiate and specify the role of BMP 5 and BMP 10 in human embryos.
Line 106: Please, add a reference.
Section 3.2 - Transforming Growth Factor-beta (TGF-β) superfamily
Lines 169-171: the sentence “In fact, TFG-β expression at porcine conceptus-maternal interface increases significantly as the concepts lengthen and the implantation process begins” is not clear. Could the author better explain this concept?
Line 178: does the author mean SMAD instead of SMAS?
Section 3.3 Fibroblast Growth Factor (FGF) family
Line 184: please, change earth with heart.
Line 190: in which steps of reproduction mechanisms the activities of FGF start?
3.4 - Insulin-like Growth Factor (IGF) system
Line 229-230, 236-238: in the lines, 229-230 the author underline that the addition of IGF-1 in culture oocyte medium did not affect blastocyst development rate in cattle but in the following paragraph, in lines 236-238 the author affirmed that addition of IGF-1 in culture medium improves blastocyst development rates in cattle. I suggest modifying properly the part to solve the contradiction.
Line 246: please, change specie with species.
3.5 - 3.6. Other growth factors
Line 289: what the author means with BSA? Please, specify.
Line 283-293: this paragraph needs to be extended with more information about the other factors affecting the oocyte and early embryonic development.
Author Response
I appreciate all the recommendations made by the reviewer and I hope that the changes made to the manuscript are to your liking. Next, I answer each one of them one by one

Reviewer 2 Report
The review “Pluripotency and growth factors in early embryonic development of mammals: a comparative approach” highlights differences in the spatial expression of the principal growth factors implied during embryo development in mammals. The manuscript reports many studies and effort was surely important. However, a careful revision should be done, regarding many aspects.
The most important missing question is that position, signaling and inherent heterogeneities all are interconnected and influence gene expression, and therefore cell fate decisions. Thus, growth factor expression is linked to transcription factor (TF) expression (and gene expression in general), and TF expression is linked to cell response, including the expression of new receptors, new signal molecules, and new TFs. So, timing is also important, before deducing differences.
Another not clear question is the scope of the review. I mean, the review is not enough educational (a true introduction is lacking ) and not enough specific. The author should decide and enhance missing parts.
Moreover, the author should pay attention to the difference between TF and GF (see 153: Another relevant group of transcription factors that are conserved across species before and during implantation is the Transforming Growth Factor beta (TGF-β) superfamily. )
Last but not least, English should be carefully revised and a more fluent writing style should be used.
Suggestions:
add a figure showing mammal embryo development (at least first stages) and different time points in different species;
add a column in each table reporting temporal expression, separated from the column reporting spatial expression.
Author Response
I appreciate all the recommendations made by the reviewer and we hope that the changes made to the manuscript are to your liking. Next, we answer each one of them one by one

Round 2
Reviewer 1 Report
I appreciated the heavy work done by the author, on this revised version of the manuscript, in which all the suggestions and corrections have been acknowledged.
Reviewer 2 Report
The manuscript was carefully revised according to my suggestion, and I have no further suggestions to make.